# A Pan-Cancer Assessment of *RB1*/*TP53* Co-Mutations

**DOI:** 10.3390/cancers14174199

**Published:** 2022-08-30

**Authors:** Ling Cai, Ralph J. DeBerardinis, Guanghua Xiao, John D. Minna, Yang Xie

**Affiliations:** 1Quantitative Biomedical Research Center, Department of Population and Data Sciences, UT Southwestern Medical Center, Dallas, TX 75390, USA; 2Children’s Research Institute, UT Southwestern Medical Center, Dallas, TX 75390, USA; 3Simmons Comprehensive Cancer Center, UT Southwestern Medical Center, Dallas, TX 75390, USA; 4Howard Hughes Medical Institute, University of Texas Southwestern Medical Center, Dallas, TX 75390, USA; 5Department of Bioinformatics, University of Texas Southwestern Medical Center, Dallas, TX 75390, USA; 6Hamon Center for Therapeutic Oncology Research, University of Texas Southwestern Medical Center, Dallas, TX 75390, USA; 7Department of Pharmacology, UT Southwestern Medical Center, Dallas, TX 75390, USA; 8Department of Internal Medicine, University of Texas Southwestern Medical Center, Dallas, TX 75390, USA

**Keywords:** *RB1*, *TP53*, neuroendocrine, sarcoma, small cell carcinoma, co-mutation, AACR-GENIE

## Abstract

**Simple Summary:**

Cancers are caused by genetic alterations called mutations. In some cases, specific mutation combinations act synergistically to provide unique advantages for cancer development. These mutation combinations are observed more frequently than by random chance. In this study, we investigated a large public tumor mutation database and found the most diverse and frequent concurrent mutations occur in *TP53* and *RB1*. We enumerated the cancer types with *TP53/RB1* co-mutations and investigated the patient outcome and the specific characteristics of cancer cells with *TP53/RB1* co-mutations, especially the drugs that can and cannot be used to kill these cells. Our work provides a tool for cancer researchers to investigate co-mutations and provides insights into the treatment of *TP53/RB1* co-mutated cancers.

**Abstract:**

Nearly all tumors have multiple mutations in cancer-causing genes. Which of these mutations act in tandem with other mutations to drive malignancy and also provide therapeutic vulnerability? To address this fundamental question, we conducted a pan-cancer screen of co-mutation enrichment (looking for two genes mutated together in the same tumor at a statistically significant rate) using the AACR-GENIE 11.0 data (AACR, Philadelphia, PA, USA). We developed a web tool for users to review results and perform ad hoc analyses. From our screen, we identified a number of such co-mutations and their associated lineages. Here, we focus on the *RB1/TP53* co-mutation, which we discovered was the most frequently observed co-mutation across diverse cancer types, with particular enrichment in small cell carcinomas, neuroendocrine carcinomas, and sarcomas. Furthermore, in many cancers with a substantial fraction of co-mutant tumors, the presence of concurrent *RB1/TP53* mutations is associated with poor clinical outcomes. From pan-cancer cell line multi-omics and functional screening datasets, we identified many targetable co-mutant-specific molecular alterations. Overall, our analyses revealed the prevalence, cancer type-specificity, clinical significance, and therapeutic vulnerabilities of the *RB1/TP53* co-mutation in the pan-cancer landscape and provide a roadmap forward for future clinical translational research.

## 1. Introduction

Self-sufficiency in growth signals was proposed as the first hallmark of cancer [1]. This property was generally thought to be driven by mutations in oncogenes, but in recent years, escaping from a dormant state by lineage plasticity has also been recognized as a hallmark of cancer [2]. Notably, lineage transdifferentiation is remarkably exemplified by small cell lung cancer (SCLC), a high-grade neuroendocrine carcinoma, with almost ubiquitous co-mutation of *RB1* and *TP53* [3]. In non-small cell lung cancer and prostate cancer, neuroendocrine lineage transition is frequently observed in treatment-resistant tumors harboring *RB1/TP53* co-mutation [4,5,6]. In normal lungs, *RB1* and *TP53* suppress the self-renewal program in pulmonary neuroendocrine cells that have stem cell potential [7], suggesting the loss of these two tumor suppressor genes can corroborate to disrupt cell fate control.

Intrigued by the cooperation of *RB1*/*TP53* co-mutations, we decided to take an in-depth co-mutation investigation with the AACR Project GENIE data. AACR-GENIE is a publicly accessible international cancer registry assembled from 19 cancer centers [8]. The latest release (v11.0) contains mutation, fusion, and copy number alteration data for over 136,000 sequenced samples from over 121,000 patients [9]. The curated samples and patient data provide information about cancer types (from 760 detailed cancer types), vital status, and sample types (metastasis, primary), etc. The scale of this dataset makes it possible to identify many statistically enriched co-mutations across different cancer types. We, therefore, performed a co-mutation screen to understand the pan-cancer landscape of co-mutations. We specifically characterized *RB1*/*TP53* co-mutations due to their abundance, diverse presence, and biological significance.

## 2. Results

### 2.1. RB1/TP53 Is One of the Most Frequently Co-Mutated Gene Pairs across Diverse Cancer Types

AACR-GENIE 11.0 data covers results from a total of 94 gene panels. We selected 76 genes that are covered by at least 50 different gene panels (Appendix A) to assess the enrichment of 2,850 combinations of co-mutations (Appendix A). *TP53* is the most frequently mutated gene with a mutation frequency of 39%, followed by *KRAS*, with a mutation frequency of 15% (Figure 1a). Although *TP53/KRAS* co-mutation ranks 1st of all co-mutation combinations, it is only significantly enriched in 5 out of 750 detailed cancer types after controlling for multiple comparisons. In contrast, *RB1* is mutated in 4.2% of the tested cases and ranks 12th on the list (Figure 1a), but by the total number of co-mutated cases, *RB1/TP53* co-mutation ranks 5th of all the co-mutation combinations, with 3832 co-mutated cases found in a total number of 128,348 cases (Figure 1b). Notably, the four co-mutations that rank before *RB1/TP53* have the majority of their cases dominated by a few cancer types. Interestingly, *RB1/TP53* is the most diversely co-mutated gene pair. It is significantly enriched in 46 out of 750 detailed cancer types (Figure 1c), suggesting its versatile functional role in cancers of different lineages.

### 2.2. A Web Tool for Performing Co-Mutation Analysis with the AACR-GENIE v11.0 Data

We constructed a web application “comut”, at https://lccl.shinyapps.io/comut/ (accessed on 1 August 2022), to allow users to review screening results and perform ad hoc analyses. To be more comprehensive, we loosened the cut-off to include 178 genes covered by at least 40 gene panels, resulting in 15,753 unique gene pairs. Note that this may, however, increase the variability of sample size and cancer types across different gene pairs, making the results less comparable. Results from concurrent and mutually exclusive co-mutations were calculated, respectively. Two sets of analyses were performed, one only considered mutation and gene fusion events, and the other added in copy number alteration (CNA) events. In the latter approach, we referred to the COSMIC Cancer Gene Census (CGC) [10] annotation of mutation types for cancer driver genes to determine the type of CNA to include. We only consider deletion as mutations when “D” is included as the mutation type from CGC, and vice versa for amplification events. However, with this approach, pairs of genes physically adjacent to each other have exceedingly high numbers of significant co-mutations across different cancer types. Such results should be interpreted with a grain of salt. In Figure 1d–f, we provide a snapshot of the web application interface. This web tool allows users to review the number of cancer types with significant co-mutation in different gene pairs to evaluate co-mutation cancer diversity (Figure 1d). It also lists the number of significantly co-mutated gene pairs by cancer types (Figure 1e). Average numbers of total cases per cancer type are provided in this table so that one can take into consideration that the power to detect statistically significant co-mutations goes down with smaller sample sizes. We also provide the results from all combinations of gene pairs and cancer types (Figure 1f), so users may derive insights by different sorting and filtering strategies, such as looking up their gene of interest to find the top co-mutated partners, or looking up their cancer type of interest to find the top co-mutated gene pairs, etc. Besides precalculated screening results, we also allow users to perform ad hoc analyses with this web tool. Users may browse the full table from co-mutation analysis for a specified gene pair (Appendix A). They may also generate an interactive scatterplot (Figure 2a).

### 2.3. RB1/TP53 Co-Mutation Is Enriched in Small Cell Carcinoma, Neuroendocrine Carcinoma, and Sarcomas

We further integrated copy number and gene fusion data to determine the concurrent loss status of *RB1/TP53* (Appendix A). With these additional annotations, *RB1/TP53* co-mutation (concurrent loss) is significantly enriched in 76 out of 576 detailed cancer types that each has at least 5 cases (Appendix A). When we review the frequency distribution of samples by *RB1/TP53* mutation status in these 76 cancer types, we found in most of these cancer types, the number of *RB1/TP53* co-mutants is much larger than the number of samples with *RB1* mutation alone whereas there are many more samples with *TP53* but not *RB1* mutation (Figure 2b). High frequencies of *RB1/TP53* co-mutation were observed for many types of small cell carcinomas besides SCLC, such as small cell bladder cancer, small cell carcinoma of unknown primary, small cell carcinoma of the stomach, etc. However, despite the high *RB1/TP53* co-mutation frequencies, due to the small sample sizes, the enrichment of *RB1/TP53* co-mutation has not reached or cannot be calculated for statistical significance in some of these cancers (Appendix A). We also observed higher frequencies of *RB1/TP53* co-mutation in many neuroendocrine carcinomas, such as uterine, prostate, head and neck, etc. neuroendocrine carcinoma. Interestingly, frequencies of *RB1/TP53* co-mutations were also found in many types of sarcomas, such as pleomorphic liposarcoma, leiomyosarcoma, myxofibrosarcoma, etc. The overall *RB1/TP53* co-mutation frequencies are significantly higher in small cell carcinoma, neuroendocrine carcinoma, and sarcomas compared to the other cancer types (Figure 2c).

### 2.4. RB1/TP53 Co-Mutants Are More Aggressive in Many Types of Cancer

The AACR-GENIE 11.0 data contains vital status but not follow-up time, so we compared the frequency of *RB1/TP53* co-mutants by vital status in different types of cancer. After controlling for multiple comparisons, there are 12 types of cancer with *RB1/TP53* co-mutation more frequent in dead patients (Figure 3a), but none of the cancer types have *RB1/TP53* co-mutation more frequent in living patients (Appendix A). Among the four oncogenotypes by *RB1* and *TP53* mutation status, the highest death rate was found in *RB1/TP53* co-mutants for almost all of these cancer types (Appendix A). When we compared the frequency of *RB1/TP53* co-mutants by sample type (primary vs. metastasis) (Appendix A), we observed many of the cancer types with co-mutants enriched in the dead patients also have co-mutants enriched in the metastatic samples, such as prostate adenocarcinoma, lung adenocarcinoma, and pancreatic adenocarcinoma. Interestingly, several types of breast cancer have co-mutants enriched in the primary cancer samples (Appendix A), despite also having co-mutants enriched in the dead patients (Figure 3a). We hence examined *RB1*/*TP53* co-mutation pattern by both vital status and sample type in these breast cancer types (Figure 3b) and found that the death association of *RB1*/*TP53* co-mutation is much more prominent in primary cancer samples than in metastatic samples, suggesting *RB1*/*TP53* co-mutation confers a more aggressive disease and account for more deaths in primary breast cancer.

### 2.5. RB1/TP53 Co-Mutation Confers Unique Therapeutic Vulnerability in Cancer

With multi-omics and high-throughput functional screening data available from a large panel of cell lines, we performed lineage-adjusted comparative analysis to identify features associated with *RB1/TP53* co-mutation (Appendix A). In the panel of cell lines previously profiled by CCLE and DepMap, very few cell lines contain only *RB1* but not *TP53* mutation, i.e., the majority of the *RB1* mutations were found concurrently with *TP53* mutations (Figure 4a), agreeing with our observation from AACR-GENIE data (Figure 2b). As a result, the comparison between RB1mutTP53wt and WT is short of statistical power, so we only examined the comparisons for *RB1/TP53* co-mutants vs. WT and RB1wtTP53mut vs. WT. We found many more significant hits from the co-mutant vs. WT comparison than the RB1wtTP53mut vs. WT comparison (Appendix A, and Figure 4b,c). From the RPPA analysis (Figure 4b and Appendix A), as expected, Rb was the most down-regulated protein in the co-mutant cell lines. Decreases in components of the Hippo pathway (YAP, TAZ) and Receptor Tyrosine Kinase (RTK) signaling (HER2, EGFR phosphorylation, IRS1) were also seen. On the other hand, cyclin E was prominently upregulated in the co-mutants as a result of E2F release from Rb suppression. Proapoptotic proteins (Bcl2 and Bim) and microtubule assembly regulators (Stathmin and acetyl-tubulin) were also found to increase in the co-mutants. These results from RPPA data connect quite well with results from the functional screening data. The co-mutants are resistant to genetic or pharmacological inhibition of upstream regulators of Rb, such as cyclin D and CDK4/6, but they have become more sensitive to inhibition of Rb downstream, such as cyclin E, E2F, CDK2, and WEE1, as well as other cell cycle regulators, such as PLK1 and aurora kinases (Figure 4c,d and Appendix A). Beyond the cell cycle, the co-mutants are sensitive to Bcl2 inhibitors and microtubule inhibitors, and resistant to inhibition of RTK, MAPK/ERK, and PI3K/mTOR pathway (Figure 4d), agreeing with the changes observed in the RPPA data. In addition, co-mutants are also more sensitive to the inhibition of HDAC and PARP, as well as various chemotherapy agents that target DNA replication (Figure 4d). These results suggest the addition of *RB1* mutation to *TP53* mutation substantially reprogrammed the cell and confers unique therapeutic resistance but also vulnerabilities that may be exploited.

## 3. Methods

### 3.1. Development of Web Application “Comut”

The web application https://lccl.shinyapps.io/comut/ is a shiny app deployed at the shinyapps.io servers. It is implemented through the following R packages: ‘shiny’, ‘data.table’, ‘ggplot2’, ‘DT’, ‘plotly’, ‘RColorBrewer’, ‘shinyjs’, ‘shinythemes’, ‘scales’, and ‘ggrepel’. Screening results and ad hoc analysis results are downloadable from the web application. The source code is available at https://github.com/cailing20/comut.

### 3.2. Assessment of Pan-Cancer Co-Mutation Frequency and Diversity

We restricted our analysis to genes covered by at least 50 panels in the AACR GENIE data. For each gene pair, only cases from platforms that cover both genes were used to assess the co-mutation frequency. Copy number data and fusion data are not integrated into the analysis for this analysis. For each cancer type, one-tailed Fisher’s exact test was used to assess the enrichment of co-mutation. The *p*-values were controlled for multiple comparisons by Benjamini–Hochberg procedures. Results with adjusted *p*-values less than 0.05 were considered statistically significant.

### 3.3. Analysis of RB1/TP53 Co-Mutation Enrichment

Based on mutation data, 128,348 cases were available from gene panels that cover both *RB1* and *TP53*. To be more comprehensive, we also utilized the discrete copy number and genomic fusion data from AACR-GENIE 11.0 to define the genomic alteration status of *RB1* and *TP53*. We referred to the COSMIC Cancer Gene Census (CGC) [10] annotation of mutation types for cancer driver genes to determine the type of CNA to include. A “loss” status was assigned to cases with copy number values less than zero or if a fusion event involves *RB1* or *TP53*. Note that this result is slightly different from the result in our web tool, as COSMIC CGC did not include “D” as a mutation type for *TP53* whereas we have considered *TP53* deletion in our analysis.

### 3.4. Association of RB1/TP53 Co-Mutation with Cancer Aggressiveness

Fisher’s exact test was used to assess the association between RB1/TP53 co-mutation (co-mutant vs. others) and patient death (alive vs. dead), or sample type (primary vs. metastasis) in different detailed cancer types. Two sets of results were obtained from one-tailed tests enforcing different directionalities. The *p*-values were controlled for multiple comparisons by Benjamini–Hochberg procedures. Results with adjusted *p*-values less than 0.05 were considered statistically significant.

### 3.5. Differential Analyses of RB1/TP53 Mutation-Associated Features in Cell Line Datasets

RPPA data was downloaded from the DepMap data portal [11]. Gene dependency datasets (CRISPR, RNAi) [12,13] and compound screening datasets (CCLE, CTRP, GDCS, PRISM primary, and secondary screens) [14,15,16,17] were downloaded and processed as previously described [18]. Briefly, the cell line names were harmonized by mapping to RRIDs, and the compound names were unified by mapping to PubChem IDs. To identify features associated with *RB1/TP53* co-mutants or RBwtTP53mut cell lines, for each feature, we fitted a linear model with lineage and binary mutation status (mutant vs. WT) as predictor variables. *p*-value and *t*-value for the mutation term were extracted and recorded. Multiple comparison-adjusted *p*-values were generated with the Benjamini–Hochberg procedures.

## 4. Discussion

Cross-cancer mutation patterns of co-occurrence and mutual exclusivity are informative of the epistatic relationship between driver genes [19]. Previous pan-cancer co-mutation analyses were mostly based on TCGA and ICGC datasets [20,21]. In this study, we performed co-mutation analyses with the AACR Project GENIE data, which has a sample size and lineage diversity ten times greater than TCGA and ICGC combined. The co-mutation screen we have conducted revealed essentially all of the important co-mutations in the most frequently mutated cancer genes across the whole spectrum of tumor lineages. We focused on *RB1/TP53* co-mutation, the most diverse and abundant co-mutation, and performed in-depth characterization with tumor and cell line datasets.

Although previous genomic studies have documented enrichment of *RB1* and *TP53* co-mutation in a few cancer types [3,4,22,23,24], a comprehensive assessment of the clinical significance of this combination across many human cancers has not been performed. We found *RB1*/*TP53* co-mutation as the most frequently observed co-mutation across diverse cancer types. However, the true frequency of *RB1/TP53* concurrent loss may be much higher. First, copy number data were not available for 32,420 of the cases, and fusion data were only available for 6 out of the 19 participating centers, so we might have underestimated *RB1* loss frequency. Second, in addition to genetic aberrations at the gene loci, several oncogenic viruses also simultaneously target p53 and pRb proteins, these include: E6 and E7 proteins from human papillomavirus (HPV), E1A and E1B proteins from adenoviruse and SV40 large T antigen [25,26]. We are unable to estimate the frequency of viral-induced *RB1/TP53* co-inactivation in our analysis. Interestingly, *RB1*/*TP53* co-mutation is especially enriched in small cell carcinomas and neuroendocrine carcinomas. Although pRb is most famous for its role in regulating cell cycle progression, it can also cooperate with differentiation-specific transcription factors for lineage specification [27,28]. Hence, the cancer lineage specificity we observed might be tied to the unique lineages that depend on pRb regulation. However, the role of p53 loss in lineage dysregulation remains elusive. Apart from small cell and neuroendocrine lineages, the frequent presence of *RB1/TP53* co-mutation in sarcomas suggests *RB1/TP53* co-mutation may not be sufficient to sustain cells in the neuroendocrine lineage. This agrees with the previous findings from a genetically engineered mouse model with *RB1/TP53*/*Myc* mutations, that depletion of the neuroendocrine lineage driver *Ascl1* led to the emergence of osteosarcoma and chondroid tumors [29].

Our analyses revealed that *RB1*/*TP53* co-mutation is associated with poor outcomes. The presence of *RB1*/*TP53* co-mutation substantially altered cancer dependencies and therapeutic vulnerabilities. As Rb loss is associated with silencing of YAP and dysregulation of the cell cycle, many of our differential vulnerability findings agree with previous studies that investigated cancer vulnerabilities defined by YAP/TEAD activity [30] or differential cell cycle utilization [31]. The acquired resistance to a wide range of targeted therapy (RTK, MAPK/ERK, PI3K/AKT) suggests *RB1* loss should be a red flag for adopting such therapies even when actionable mutations are present in the targets. This is already known for the case of lung adenocarcinoma with EGFR mutations [32]. On the other hand, the acquired susceptibility to a wide range of compounds as well as chemotherapies that target DNA replication suggests loss of *RB1* could potentially be considered actionable for targeted therapies. However, SCLC, the cancer type with the highest frequency of *RB1/TP53* co-mutations, is well known for its good initial response to chemotherapy and notorious for its inevitable relapse. It remains to be determined whether similar refractory phenomena and mechanisms exist for *RB1/TP53* co-mutants in other cancer types.

## 5. Conclusions

Overall, our web tool provides a resource for use by the field and future study on pan-cancer co-mutation landscapes. Our in-depth investigation of *RB1*/*TP53* co-mutations also provides a roadmap forward for treating cancers with this aggressive mutation combination.

## Figures and Tables

**Figure 1 cancers-14-04199-f001:**
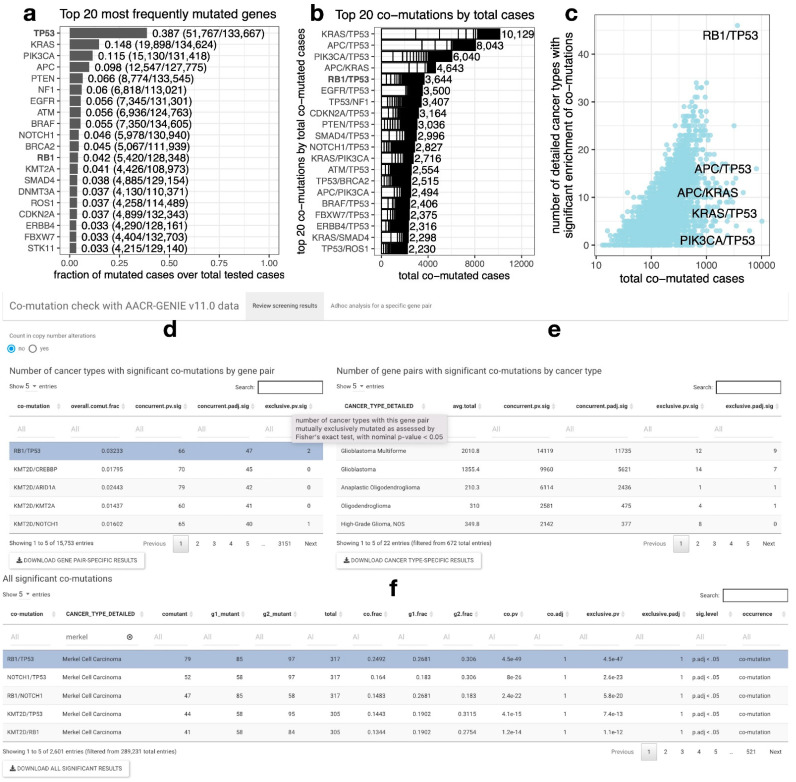
*RB1/TP53* co-mutation ranks top by the number of cases and the number of cancer types. (**a**). Top 20 most frequently mutated genes from 76 cancer genes covered by at least 50 gene panels. (**b**). Top 20 gene pairs by total co-mutated cases. To show the cancer diversity for each co-mutation combination, case contribution from different cancer types is visualized as stacked grids within each bar and each grid represents a contribution from one cancer type. Note that for the top four mutation pairs, the majority of the cases were contributed by a small number of cancer types. (**c**). *RB1*/*TP53* co-mutation is the most diversely co-mutated pair from 2850 pairs of co-mutations screened. One-tailed Fisher’s exact test was used to determine the statistical significance of co-mutation enrichment in about 700 detailed cancer types. The number of cancer types with significant enrichment in specific co-mutations is shown on the *y*-axis, and the number of total co-mutated cases is shown on the *y*-axis. (**d**–**f**). Screening result review from the “comut” web application. The number of cancer types with significant co-mutation for a given gene pair can be reviewed from the upper left table (**d**). The number of gene pairs with significant co-mutations for a given cancer type can be reviewed from the upper right table (**e**). All the significant results for different combinations of gene pairs and cancer types can be reviewed from the bottom table (**f**). These tables are filterable and downloadable. The full name and explanation of the table headers are provided in a tooltip upon mouseover. For example, in this snapshot, a tooltip for “exclusive.pv.sig” is displayed. Abbreviated for headers: overall.comut.frac, the fraction of co-mutated samples from all cancer types; *.sig, number of significant hits by nominal (*.pv.sig) or adjusted (*.padj.sig) *p*-values in tests for concurrent or mutually exclusive co-mutation; avg.total, averaged total cases belonging to the specific cancer type, the total cases vary by gene pair due to variable coverage of gene panels; comutant, number of co-mutated cases; g1_mutant and g2_mutant, number of cases with mutations on gene 1 and gene 2 (left and right part of “co-mutation”); total, the total number of cases; co.frac, g1.frac, g2.frac are comutant, g1_mutant, g2_mutant divided by total. “*” is a wildcard that represents “concurrent” or “exclusive”.

**Figure 2 cancers-14-04199-f002:**
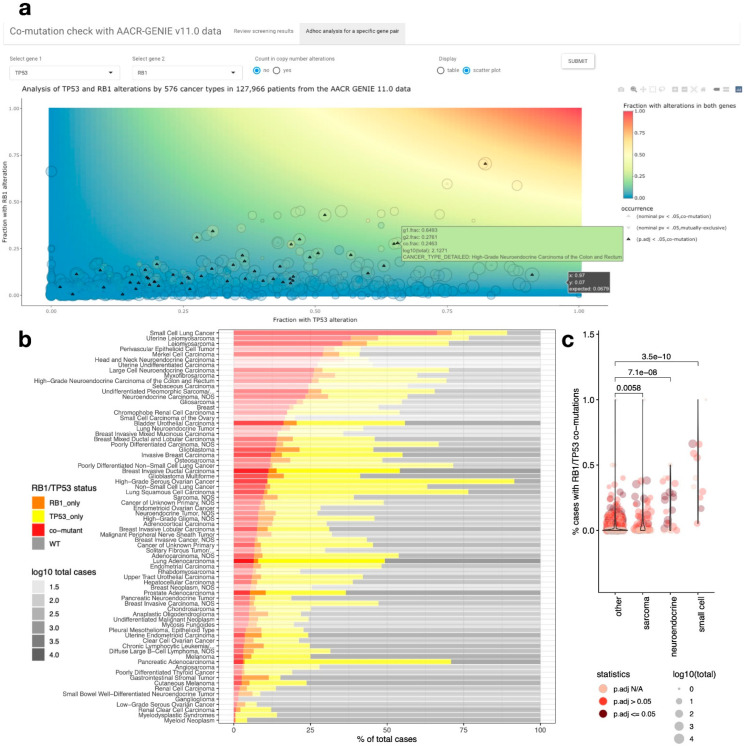
Pan-cancer landscape of RB1 and TP53 mutations. (**a**). Screenshot from “comut” web tool for Adhoc co-mutation analysis. Users can specify a gene pair, and whether to include copy number alterations in mutation for analysis. The interactive scatterplot allows users to inspect results from different cancer types upon mouseover. The background color of the plot indicates the expected co-mutation frequency assuming independence of each mutation. (**b**). Cancer types with significantly enriched *RB1*/*TP53* co-mutations. Fraction of cases with *RB1*/*TP53* co-mutation, *RB1* mutation only, *TP53* mutation only, or WT in both genes were visualized as stacked bar plots. The transparency of the bars denotes the number of cases for each cancer type. Full name for “Undifferentiated Pleomorphic Sarcoma/…” is “Undifferentiated Pleomorphic Sarcoma/Malignant Fibrous Histiocytoma/High-Grade Spindle Cell Sarcoma”, full name for “Solitary Fibrous Tumor/…” is “Solitary Fibrous Tumor/Hemangiopericytoma”, and full name for “Chronic Lymphocytic Leukemia/…” is “Chronic Lymphocytic Leukemia/Small Lymphocytic Lymphoma”. (**c**). *RB1*/*TP53* co-mutation frequencies are higher in small cell carcinoma, neuroendocrine carcinoma, and sarcoma. Individual cancer types were represented by jittered points in the category denoted on the x-axis label. Statistical significance of *RB1*/*TP53* co-mutation enrichment in each cancer type was denoted by color and the total number of cases was denoted by point size. The overall distribution of each category is visualized as violin plots. *p*-values for pairwise comparison between “other” and selected cancer groups were calculated by the Mann-Whitney test.

**Figure 3 cancers-14-04199-f003:**
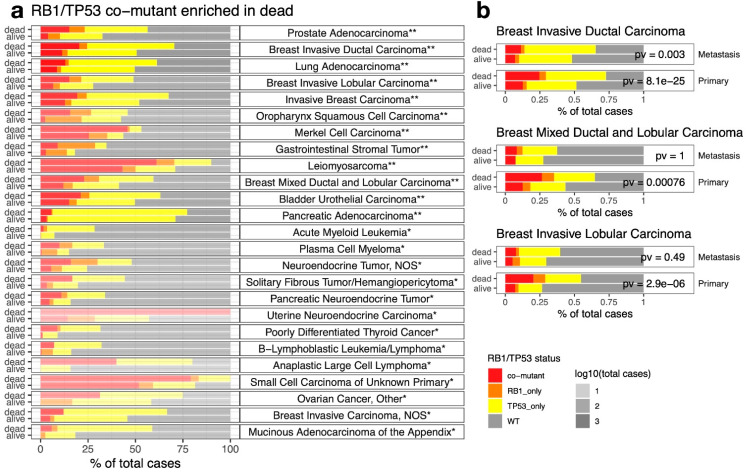
Association between *RB1*/*TP53* co-mutations and patient death. (**a**). *RB1*/*TP53* co-mutations are more are frequent in dead than in alive patients for many cancer types. Cancer types are ordered by *p*-values. The distribution of samples by *RB1*/*TP53* co-mutation status was visualized as stacked bar plots for 25 cancer types with higher co-mutation frequency found in the dead patients. The transparency of the bars denotes the number of cases for each cancer type. *p*-values were denoted by asterisks (**, adjusted *p*-value < 0.05, *, nominal *p*-value < 0.05) or printed on the graph. (**b**). Enrichment of *RB1*/*TP53* co-mutations in dead patients is more prominent in primary than in metastatic breast cancer. *p*-values from Fisher’s exact tests are provided.

**Figure 4 cancers-14-04199-f004:**
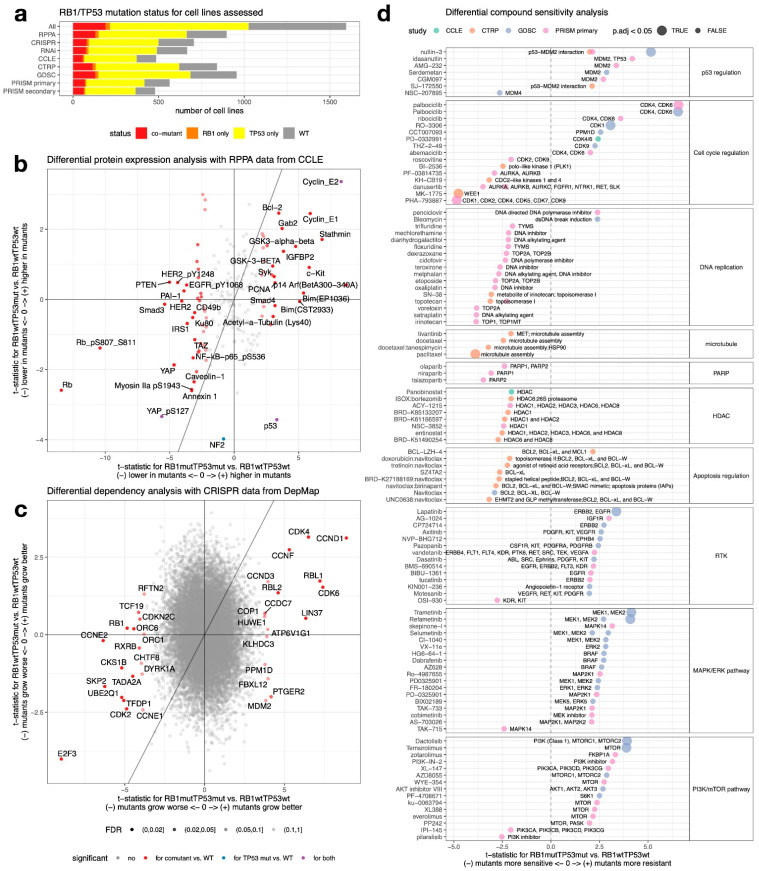
Therapeutic vulnerability of *RB1*/*TP53* co-mutants. (**a**). *RB1*/*TP53* mutation status in cell lines datasets used for association analyses. (**b**–**d**), Identification of differential features from RPPA functional proteomics data (**b**), CRISPR gene dependency data (**c**), and four sets of compound screening data (**d**). Lineage-adjusted linear models were used to generate t-statistics comparing mutants against WTs (reference). In the scatterplots (**b**,**c**), *x*-axis values are from comparing co-mutant and WT whereas *y*-axis values are from comparing RB1wtTP53mut and WT, line x = y is added. Hits with a false discovery rate (FDR) < 10% are colored and labeled in **c** whereas only hits with FDR < 2% are labeled in (**b**) due to limited space. In the analysis of drug sensitivity (**d**), compounds were categorized by the pathways they act upon. Within each category, *t*-values for compounds with nominal *p*-value < 0.05 were plotted, with colors indicating data source, sizes indicating significance after controlling for multiple comparisons, and labels indicating drug targets.

## Data Availability

AACR GENIE v11.0 data was downloaded using Synapse Python Client with accession id syn26706564 on 5 April 2022. The processed dependency and compound screening datasets for cell lines can be downloaded from our previously published web app https://lccl.shinyapps.io/FDCE/.

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
