# Peer review of "A Pan-Cancer Assessment of RB1/TP53 Co-Mutations"

_cancers, 2022, doi:10.3390/cancers14174199_

Round 1

Reviewer 1 Report

This paper investigated the top co-mutation from pan-cancer mutation datasets and omics datasets. They focused on the top comut RB1/TP35 pair and explored clinical significance, and therapeutic vulnerabilities to the comut pair. The group also built a light shiny app for users to explore and filter the comut results. The author did a great job in pan-cancer mutation/omics datasets integration and performing comprehensive visualizations. The statistical and bio-informatical methods need to be discussed in more detail. The writing is cohesively reasoned but the language of the text is confusing in some places. Several major improvements need to be made for publication.

Major:

1.    What would be the rationale for selecting genes shown in more than 50 gene panels? The author stated that the data used in the shiny app is based on a different cut-off. It seems arbitrary to select genes that are suitable for the analysis.

2.    The high incidence of RB1/TP53 in the small cell cancer family is intriguing, would the author extend the analysis and discussion for pathological implications? Does RB1/TP53 co-loss in small cell cancers elevate compared with another tumor family? It would be interesting to see whether this elevation is present in cell line data with small cell tumor origins.

3.    Is Fig.3a ranked by P value? The plot is too busy for visualization and needs further abstraction. Could the author only display the P value of the difference between live and dead patients, and group the tumor families?

4.    The screenshots from the app have low resolution and small font sizes that are not suitable for publication purposes.  Given the mostly static pre-computed data inputs, the majority of the function of the app can be achieved by SQL query.  More visualization and functions could be built into the app for example incorporating the omics data and the clinical status annotations.

Minor:

1.    Figure1b, it may be more clear if the author could add numbers of cancer types for each co-mutation pair.

2.    Some cancer type names in Fig. 2a are not complete.

3.    The abbreviations in attribute names in the shiny app are not interpretable, it would be great to add the full name, and a description of how each attribute is calculated.   

4.    Many bio-informatics analysis of the method sections is not precise. It would be great if the author would give more details on the omics data analysis pipeline.

Author Response

Thank you for your commends and suggestions. Please see our rebuttal letter in pdf.

Reviewer 2 Report

Two major limitations:

1.     Co-mutation analyses tools already exist; nothing new in this study:

https://pubmed.ncbi.nlm.nih.gov/30629309/ and https://bmcmedgenomics.biomedcentral.com/articles/10.1186/s12920-019-0510-y

Similar analyses could be performed using tools at

https://cancer.sanger.ac.uk/cosmic and https://academic.oup.com/nar/article/49/D1/D1289/5976976

2.     Co-mutations described in this study have already been described and catalogues, some examples:

https://www.ncbi.nlm.nih.gov/pmc/articles/PMC8774165

https://www.ncbi.nlm.nih.gov/pmc/articles/PMC7043073

https://www.cell.com/trends/cancer/fulltext/S2405-8033(21)00101-1

Publicly available information at NCI not referred to:

https://www.cancer.gov/research/key-initiatives/ras/ras-central/blog/2015/kras-cancer-co-mutation

In the light of the above two limitations, the work is unsuitable for publication unless the authors describe unique novel features in their tool.

Additionally

The tool would benefit from working on newly acquired cancer genome datasets. If the tool can take novel sequence datasets as input and give co-mutated genes as output, it would be a value addition making it suitable for publication.

As it is, manuscript is not novel and adds no value to the existing knowledge in this field. 

Author Response

(The authors gave the same response as above.)

Reviewer 3 Report

The authors have put sincere efforts into this manuscript. The study provides an interphase for quick analysis of co-mutation in cancers and conclusions on the RB1/TP53 co-mutation in many different types of cancers. I do have some comments to share after reading the paper and hopefully, they are helpful in further improving it.

First, the authors analyzed AACR-GENIE 11.0 data and concluded that RB1/TP53 is the most common co-mutation across different cancer types. Single mutation of RB1 is not at the top 1 or 2, and RB1/TP53 co-mutation ranks 5th in the co-mutation list. While the authors clearly explained the logic on studying RB1/TP53, the other potential top candidates are excluded. More discussions on the previous studies of single mutation RB1, TP53, and the top 1 co-mutation KRAS/TP53 will further help convince the readers. The logic to study co-mutation instead of single, triple or multiple mutations can also be further elucidated.

Second, the authors built an interphase for the users to easily access the co-mutation analysis. This is a user-friendly interphase and with trials, the website can provide results with either CNV or without CNV analysis and the results are downloadable. This will help other researchers have a quick view of the co-mutational information while counting different cancer types.

The authors concluded that the RB1/TP53 co-mutation is more enriched in three types of carcinomas. While the results are interesting with potential further study directions for other groups in the field, the authors can further emphasize the possible reasons of RB1 and TP53 co-mutation enrichment in these three cancer types. With the other previous experimental reports, the discussion on this conclusion is more reliable.

In the last part, the authors also concluded that RB1/TP5 co-mutation were found in more aggressive cancers. I concern about the single TP53 mutation counts top in most of the cases (Fig 3). Following this point, I think the authors could make a stronger case for their findings with the discussion and comparison of single TP53 mutation in different cancer types. Same for the therapeutic vulnerability study.

Overall, this study is well performed and the general web interphase can provide other researchers an overview of the co-mutation information to focus on the top ones. The aggressive RB1/TP53 co-mutation can be further investigated for different cancer types as potentially general treatment targets. I hope the comments could be helpful and I give my best wishes for the authors in publishing this study.

Author Response

(The authors gave the same response as above.)

Round 2

Reviewer 1 Report

The authors successfully addressed all of the reviewers' comments. Overall, the manuscript meets the publication requirement.

Reviewer 2 Report

The authors have satisfactory answered all the queries raised and have included the references suggested. This addition has significantly improved the quality of the paper and also emphasized the novelty of their work.